# DeepPointMap2:
# Accurate and Robust LiDAR-Visual SLAM with Neural Descriptors

## ABSTRACT

Simultaneous Localization and Mapping (SLAM) plays a pivotal role in autonomous driving and robotics. Given the complexity of road environments, there is a growing research emphasis on developing robust and accurate multi-modal SLAM systems. Existing methods often rely on hand-craft feature extraction and cross-modal fusion techniques, resulting in limited feature representation capability and reduced flexibility and robustness. To address this challenge, we introduce **DeepPointMap2**, a novel learning-based LiDAR-Visual SLAM architecture that leverages neural descriptors to tackle multiple SLAM sub-tasks in a unified manner. Our approach employs neural networks to extract multi-modal feature tokens, which are then adaptively fused by the *Visual-Point Fusion Module* to generate sparse neural 3D descriptors, ensuring precise localization and robust performance. As a pioneering work, our method achieves *state-of-the-art* localization performance among various Visual-based, LiDAR-based, and Visual-LiDAR-based methods in widely used benchmarks, as shown in the experiment results. Furthermore, the approach proves to be robust in scenarios involving camera failure and LiDAR obstruction.

## CCS CONCEPTS

• **Computing methodologies** → **Neural networks**; **Vision for robotics**; **Reconstruction**; *Scene understanding*.

## KEYWORDS

Visual-LiDAR SLAM, Multi-modal Fusion, Neural Descriptors

## 1 INTRODUCTION

Simultaneous Localization and Mapping (SLAM), aiming to estimate the agent's location while mapping its environment, is pivotal in autonomous driving and robotics, enabling navigation in unseen environments and understanding the surroundings. The environments in which autonomous vehicles operate are highly complex, making it challenging to achieve accurate SLAM with a single sensor. Multi-modal perception emerges as a crucial strategy, integrating various sensors like monocular cameras and LiDARs to enhance SLAM system capabilities.

As shown in Fig. 1, autonomous driving scenarios sometimes face challenges such as obstructed LiDAR point clouds and under/overexposure images. Such conditions can lead to a degradation

*ACM MM, 2024, Melbourne, Australia*
© 2024 Copyright held by the owner/author(s). Publication rights licensed to ACM.
ACM ISBN 978-x-xxxx-xxxx-x/YY/MM
https://doi.org/10.1145/nnnnnnn.nnnnnnn

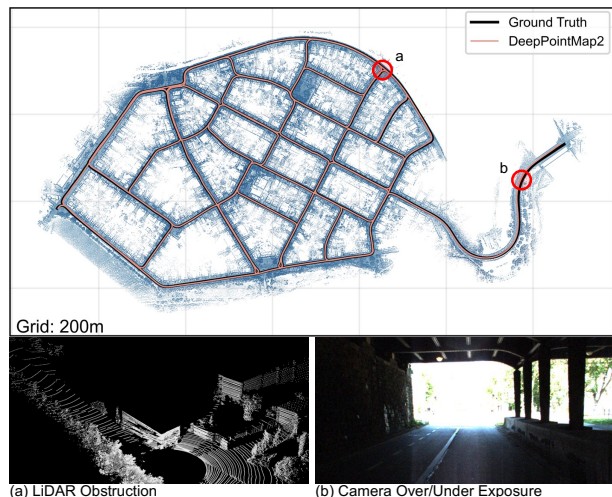

**Figure 1: Examples of challenging scenes in road environments. (a) LiDAR obstructed by large vehicles, resulting in incomplete point clouds. (b) Camera with limited dynamic range struggles to properly expose in high-contrast scenes.**

in the performance of multi-modal SLAM methods. Some existing approaches [46, 59] typically process point cloud and image data separately in individual branches and employ post-fusion to estimate the trajectory. However, these approaches suffer from a limitation in adequately integrating information from multiple modalities during the feature extraction process, leading to a weak feature representation capability. While some other approaches [4, 31, 44, 55] fuse the feature before position estimation. Although these methods achieved accurate localization performance, they lack robustness against sensor failure or degeneration.

The pioneering work, DeepPointMap [62], utilizes neural descriptors to achieve accurate localization. It is confirmed that the neural descriptors enable efficient and accurate feature representation and benefit the SLAM performance. However, it only involves LiDAR point cloud modality and encounters challenges in scenarios with sparse reference objects and limited geometric information. To address these challenges, we introduce a learning-based LiDAR-Visual SLAM approach, *DeepPointMap2*, aiming to enhance the key processes of feature extraction and localization in SLAM task using neural networks. We utilize two arbitrary backbone networks to extract multi-scale tokens from both image and point cloud. A *Visual-Point Fusion module* is followed to aggregate these multi-modal tokens into 3D neural descriptors. This end-to-end learnable framework allows the model to adaptively learn how to fuse multi-modal features, providing better feature representation and robustness, compared to existing manual-designed extraction and fusion strategies.

The proposed framework offers several advantages over existing methods. **Representation capacity**: Our method aggregates multi-scale features in a learning-based manner, providing descriptors with both local fine-grained geometric features and coarse-grained scene information, thereby enhancing the representation capacity. **Robustness**: Our approach has strong robustness by utilizing learning-based cross-modal feature fusion strategy. When encountering LiDAR obstruction or camera failure, the model can still maintain accurate localization and mapping. **Flexibility**: Our method is a simple and end-to-end pipeline, and can accommodate arbitrary single-modal backbone. Users can choose the appropriate backbone according to their needs. In addition, the network structure is also potentially able to fuse more modal inputs.

**Our contributions can be summarized as follows:**

- We proposed a pioneering learning-based Visual-LiDAR SLAM framework, *DeepPointMap2*, that leverages multi-modal neural 3D descriptors to represent the environment with high fidelity. These descriptors are designed to capture more refined features and adeptly integrate information from multiple modalities, thereby ensuring enhanced representation and robust performance.
- Our framework includes a specialized deep neural network module for robust cross-modal feature fusion, which integrates features from both images and point clouds. This module incorporates learnable weights and spatial correspondence mechanisms to fuse information effectively.
- Experimental results show that our approach achieves *state-of-the-art* (SOTA) performance in localization accuracy. Notably, *DeepPointMap2* achieves stable performance when facing modalities missing scenarios, such as in the event of camera failure or LiDAR obstruction.

## 2 RELATED WORK

### 2.1 LiDAR SLAM

LiDAR point clouds are commonly represented as unordered sets of 3D coordinates. In SLAM methods, the process typically involves first (1) extracting key-points with geometric features from the point cloud, followed by (2) matching corresponding key-points between adjacent frames based on their geometric features, and finally (3) solving the relative pose transformation through SVD [2] or iterative techniques [29]. Extracting geometric features from the point cloud constitutes a critical step in LiDAR-based SLAM approaches.

**Knowledge-based** methods compute the feature based on predefined geometric metrics, such as curvature and density. LOAM [58], as one of the early works, and its subsequent approaches [19, 42, 52, 53] utilized point-wise curvature to detect edge and planar points. Then, the association is applied within each category. Additionally, MULLS [34] further classify key-points into more specific categories to establish a more accurate association. PUMA [49] introduced a surface mesh representation that better captured the geometric appearance of objects in the scene. Although these methods can extract features efficiently, their representation capability is limited, necessitating more key-points and complex association algorithms.

**Learning-based** methods utilize deep neural networks to extract point cloud features. PointNetLK [1] employs PointNet [35] to extract scan-level features and applies a modified Lucas-Kanade algorithm for transformation estimation. LO-Net [22] proposed a scan-to-scan odometry network that predicts normals, identifies dynamic regions, and incorporates a spatiotemporal geometrical consistency constraint for improved interactions between sequential scans. To achieve accuracy loop detection, LCDNet [5] and its lightweight variant DeLightLCD [54] utilized a 3D voxel CNN network to extract descriptors and estimate coarse transformation. DeepPointMap [62] pioneers the use of neural networks for unified odometry and loop detection, employing neural descriptors for accurate localization with efficient memory use.

Despite the robustness of the LiDAR sensor, its inability to capture informative texture makes it suffer from structure-less environments (*e.g.*, tunnels). The partial obstruction issues may further hinder its performance.

### 2.2 Visual SLAM

Similar to LiDAR SLAM, monocular visual SLAM can also be considered a data association task. Visual SLAM can be categorized into indirect and direct methods depending on the association method.

**Direct** methods directly minimize the pixel-wise photometric error between frames to estimate camera motion. DTAM [33] utilized all pixels of frames and estimated the relative pose of the camera. To reduce computational complexity, LSD-SLAM [11] and DSO [10] selected pixels with large gradients. As one of the most famous methods, SVO [13] further introduced the FAST feature detector to enhance feature extraction ability and achieve precise association.

**Indirect** methods focus on detecting and matching sparse feature descriptors from images to reduce computational complexity. To achieve this, some methods [20, 21, 41, 45] extract point features from image using pixels' neighbor, while some [50, 61] focus on line features. Meanwhile, some methods utilize neural networks to select and extract descriptors. SemanticFusion [30] utilizes CNNs to perform semantic segmentation to build the semantic map. To suppress the effects of dynamic objects, Cheng et al. [6] also uses a neural network to select static sparse descriptors.

Although visual SLAM methods only require inexpensive cameras, they may encounter challenges such as sensitivity to illumination (*e.g.*, HDR environment) or weather (*e.g.*, rain).

### 2.3 Visual-LiDAR SLAM

Visual-LiDAR SLAM models can be divided into two categories, depending on the cross-modal feature fusion strategy.

**Loosely-Coupled** methods consider the estimation of several modalities separately. The cross-modal fusion procedure is applied after each estimation is generated. FAST-LIVO [63], R$^2$LIVE [26], and its subsequent R$^3$LIVE [25] employ a Kalman-Filter to fuse LiDAR, Visual, and IMU measurements, yielding precise odometry results. Similarly, LIV-LAM [39] proposes an unsupervised learning method for object discovery based on a camera detector and a LiDAR odometry, followed by the fusion of detected objects and LiDAR measurements using pose-graph optimization.

**Tightly-Coupled** methods, unlike loosely-coupled ones, fuse sensor measurements from each modality before the state estimating,

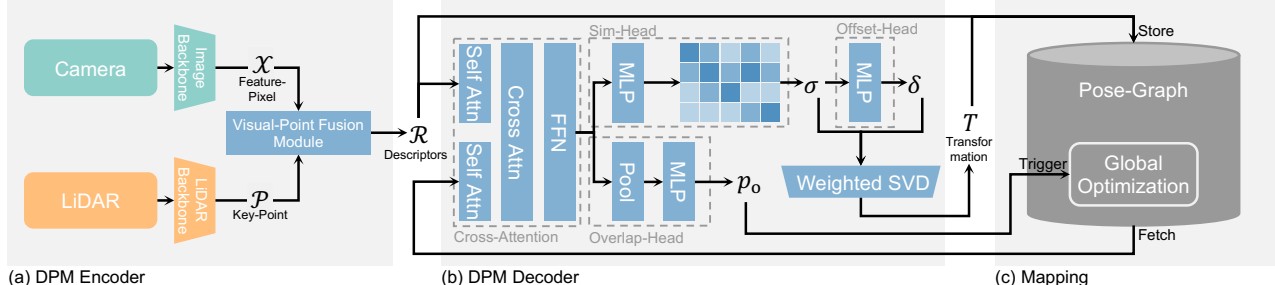

(a) DPM Encoder  (b) DPM Decoder  (c) Mapping

**Figure 2:** *DeepPointMap2* **consists of three components. (a) DPM encoder extract multi-modal features and aggregate them into 3D neural descriptors. (b) DPM decoder utilizes descriptors for solving multiple SLAM subtasks** *i.e.,* **odometry and loop detection. (3) Mapping Module manage the observation and reconstructed map.**

which are often more accurate. DEMO [57] utilized depth information from LiDAR (or RGB-D camera) to enhance the bundle adjustment-based visual odometry. In DEMO, the depth information is integrated into the optimization process to refine the estimated camera poses. Huang et al. [18] introduced a visual-LiDAR odometry method using point and line features extracted from images. After estimating depth based on LiDAR data, the point and line depths are utilized as prior factors in the point-line bundle adjustment process. LAMV-SLAM [55] integrates LiDAR and monocular visual data by employing online photometric calibration and a depth fusion algorithm to provide accurate depth values for visual features, enhancing mapping and localization in outdoor environments.

Loosely-coupled methods provide modal flexibility but risk losing critical information and accuracy due to underutilized inter-modal relationships. Although tightly-coupled methods are more integrated, they rely on all sensors operating correctly and a single sensor failure can diminish performance or cause system failure.

## 3 ARCHITECTURE

### 3.1 Model Overview

As illustrated in Fig. 2, our proposed *DeepPointMap2* consists of three main components: (a) **DPM Encoder** aims to extract feature tokens from multi-modal sensor data and aggregate them into comprehensive descriptors for each frame, (b) **DPM Decoder** utilize neural networks to estimate the transformation matrix between two frames based on their descriptors and perform loop detection to assist with constructing a consistent map, and (c) **Mapping Module** store the frame-wise information into a pose-graph structure and executes global-optimization once a loop closure is confirmed.

### 3.2 DPM Encoder

*DeepPointMap2* is a multi-modal SLAM framework that utilizes neural descriptors $\mathcal{R}$ with compressed semantic features to represent 3D scenes. A descriptor $\mathbf{r}_i$ can be denoted as $\mathbf{r}_i = (\mathbf{r}_i^{\text{xyz}}, \mathbf{r}_i^{\text{feat}})$, where $\mathbf{r}_i^{\text{xyz}}$ denotes the 3D coordinate and $\mathbf{r}_i^{\text{feat}}$ is the associated feature.

DPM Encoder takes both point cloud and image as input, as illustrated in Fig. 3 (a). For point clouds, we use PointNeXt [36], one of the most famous neural architectures for point cloud understanding, as our backbone to extract multi-scale key-points $\mathbf{p}_i := (\mathbf{p}_i^{\text{xyz}}, \mathbf{p}_i^{\text{feat}})$, where $\mathbf{p}_i^{\text{xyz}}$ is the coordinate of key-point and $\mathbf{p}_i^{\text{feat}}$ is its feature. For

image data, we employ ConvNeXt [28] with FPN [27] to extract multi-scale feature map. We denote each feature-pixel as $\mathbf{x}_i := (\mathbf{x}_i^{\text{uv}}, \mathbf{x}_i^{\text{feat}})$, where $\mathbf{x}_i^{\text{uv}}$ is its UV coordinate.

After extracting multi-scale feature tokens (as $\mathcal{P}$ and $\mathcal{X}$), the *Visual-Point Fusion Module* fuse these features and generate the descriptors $\mathcal{R}$. As shown in Fig. 3, the *Visual-Point Fusion Module* consists of two parts: Reference-Points Generator (RPG) and Multi-Modal Transformer (MMT) Decoders. Given the multi-scale tokens of both modalities ($\mathcal{X}, \mathcal{P}$), the RPG generates a set of 3D reference points (ref-points) based on the input point cloud. Subsequently, the MMT cascadely fuses feature tokens into these ref-points, and finally generates descriptors $\mathcal{R}$.

*3.2.1 Reference-Points Generator.* The Reference-Points Generator (RPG) aims to generate a set of 3D reference points (ref-points) $\mathbf{r}_i$ that serve as *seeds* for aggregating multi-modal tokens in subsequent modules. These ref-points are carefully selected to be both uniform across the point cloud and representative of the underlying scene structure, which is essential for effective aggregation and robust feature representation. To achieve this, we employ farthest-point-sampling (FPS) to select $n$ ref-points coordinates $\mathbf{r}_i^{\text{xyz}}$ from the original point cloud. This strategy ensures that the selected ref-points are well-distributed and cover the spatial extent of the environment. We then initialize the features of each ref-point using an MLP based on their spatial coordinates:

$$\mathbf{r}_i^{\text{feat}} = \text{MLP}\left(\mathbf{r}_i^{\text{xyz}}\right) \tag{1}$$

*3.2.2 Multi-Modal Transformer Decoder Layers.* A sequence of $L$ Multi-Modal Transformer (MMT) Decoders is applied to further process and refine the aggregated multi-modal tokens, following the Reference-Points Generator (RPG). Each MMT takes multi-scale (1) image feature-pixels $\mathcal{X} = (\mathbf{x}^{\text{uv}}, \mathbf{x}^{\text{feat}})$, (2) LiDAR key-points $\mathcal{P} = (\mathbf{p}^{\text{xyz}}, \mathbf{p}^{\text{feat}})$, and (3) reference points $\mathcal{R} = (\mathbf{r}^{\text{xyz}}, \mathbf{r}^{\text{feat}})$ as inputs. Initially, the ref-points are input into a multi-head self-attention module, which effectively enhances their features by allowing information exchange among ref-points. As shown in Figure Fig. 3 (b), each MMT layer consists of two branches: LiDAR and Image. Each branch is a transformer decoder structure with multiple layers, where Query is ref-points $\mathcal{R}$, both Key and Value are feature-pixels $\mathcal{X}$ in image branch or key-points $\mathcal{P}$ in LiDAR branch.

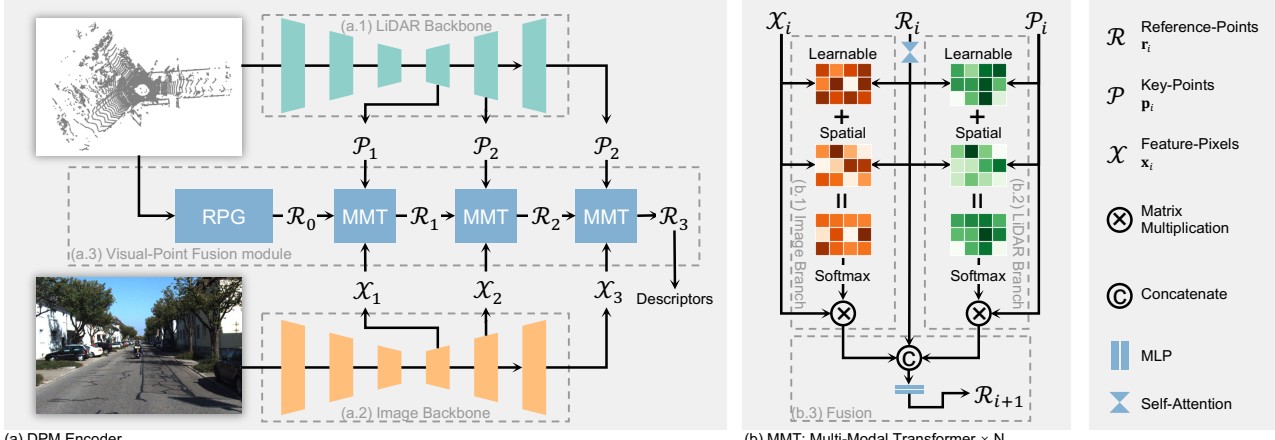

(a) DPM Encoder

(b) MMT: Multi-Modal Transformer × N

**Figure 3: Overview of DPM Encoder. The encoder takes image and point cloud as inputs, and fuses the multi-scale feature tokens into descriptors to represent the environments and solve subsequent SLAM subtasks.**

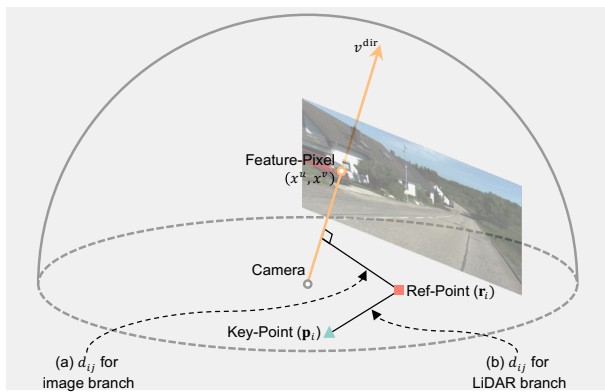

**Figure 4: Distance of Feature Tokens to Reference points.**

To further improve the performance of decoder layers, we introduce a novel cross-attention transformer module, named Biased Transformer, to replace the original scale-dot transformer block. As illustrated in Fig. 3 (c), the Biased Transformer module adaptively combines relative spatial distances with feature similarities between ref-points and feature-tokens. This integration facilitates a more precise and context-aware feature fusion strategy, thereby enhancing the overall accuracy and robustness of the SLAM framework.

**LiDAR branch:** Unlike the original scale-dot transformer [47], the Biased Transformer incorporates the distance between query and key in both spatial and feature space. Following scale-dot attention, we calculate the *learnable attention map* $E^{(1)}$ in feature space using pair-wise dot-product without positional embedding:

$$E_{ij}^{(1)} = \frac{1}{\sqrt{d}} \cdot W_q(\mathbf{p}_i^{\text{feat}}) \cdot W_k(\mathbf{r}_j^{\text{feat}}) \tag{2}$$

where $W_q$ and $W_k$ are learnable weights layers and $d$ is the feature dimension. Meanwhile, as illustrated in Fig. 4 (a), since we know the relative position between key-points $\mathbf{p}_i^{\text{xyz}}$ and ref-points $\mathbf{r}_j^{\text{xyz}}$, a

*spatial attention map* $E^{(2)}$ is derived based on Euclidean distance:

$$E_{ij}^{(2)} = \exp(-\alpha_1 \cdot d_{ij}) \tag{3}$$

where $d_{ij} = \|\mathbf{p}_i^{\text{xyz}} - \mathbf{r}_j^{\text{xyz}}\|_2$ is the distance between ref-points and key-points and $\alpha_1$ is a pre-defined hyperparameter.

The final attention map for the Biased Transformer is obtained by merging these two attention maps by:

$$E^{\text{LiDAR}} = \text{Softmax}\left(E^{(1)} + \xi\left(E^{(2)}\right)\right) \tag{4}$$

where $\xi$ is a Z-score normalization function. This normalization ensures the stability and comparability of the combined attention weights, allowing for a more effective fusion of spatial and feature information in the LiDAR branch.

**Image branch:** The *learnable attention map* $E^{(1)}$ can be calculated same as Eq. (2). However, due to the distinct coordinates of feature-pixels $\mathbf{x}_i$ and ref-points $\mathbf{r}_j$, the *spatial attention map* $E^{(2)}$ cannot be directly computed using Equation Eq. (3). Some methods project Li-DAR key-points onto image coordinates and then extract fixed-size image patch features. These patch-wise feature are attached to the corresponding key-points. However, it is important to note that due to the potential scale uncertainty introduced by perspective projection, the fixed-size fusion mechanism may be inaccurate. To solve this problem, we introduce a metric based on the point-ray distance. As demonstrated in Fig. 4 (b), each feature-pixel $\mathbf{x}_i$ in the image can be defined as a ray $\mathbf{v}_i$ emanating from the camera's optical center $\mathbf{v}_i^{\text{ori}}$, with its directional vector $\mathbf{v}_i^{\text{dir}}$ calculated from its pixel coordinate $\mathbf{x}_i^{\text{uv}}$.

$$v_i := \begin{cases} \mathbf{v}_i^{\text{ori}} = T \\ \mathbf{v}_i^{\text{dir}} = T \times K^{-1} \times (\mathbf{x}_i^{\text{u}}, \mathbf{x}_i^{\text{u}}, 1, 1)^\top \end{cases} \tag{5}$$

where $T$ is the transformation matrix from the camera to LiDAR and $K$ is the intrinsic matrix of the camera. Given the disparate sensing ranges of cameras and LIDAR, it is necessary to initially determine if each ref-point falls within the camera's field of view

and subsequently create a corresponding mask $M$ by:

$$M_{ij} = \begin{cases} 1 & \text{,if } \mathbf{v}_i^{\text{ori}} \cdot \left( \mathbf{r}_j^{\text{xyz}} - \mathbf{v}_i^{\text{ori}} \right) \geq 0 \\ 0 & \text{,otherwise.} \end{cases} \tag{6}$$

The attention map can be calculated based on the distance $d_{ij}$ between ref-points to the rays, given by:

$$d_{ij} = \frac{\mathbf{v}_i^{\text{dir}}}{\left\| \mathbf{v}_i^{\text{dir}} \right\|_2} \cdot \left( \mathbf{r}_j^{\text{xyz}} - \mathbf{v}_i^{\text{ori}} \right) \tag{7}$$

The *spatial attention map* is then calculated similar to Eq. (3) by:

$$E_{ij}^{(2)} = M_{ij} \cdot \exp\left(-\alpha_2 \cdot d_{ij}\right) \tag{8}$$

To this end, the *spatial attention map* $E_{ij}^{(2)}$ of the image branch is calculated, but now indicating the spatial relationship between the 3D ref-points and the 2D feature-pixels. It is worth noting that since the computation is done in 3D space, the weights $E_{ij}^{(2)}$ are able to precisely account for scale variations in the image that arise from perspective projection. Finally, the combined attention map $E^{\text{Image}}$ for the image branch is computed using Eq. (4), adaptively integrating both the learnable and spatial attention mechanisms.

**Feature fusion:** After obtaining the corresponding modalities features, the feature of $l$-th layer ref-points $\mathcal{R}_l^{\text{feat}}$ are updated as:

$$\mathcal{R}_l^{\text{feat}} = \text{MLP}\left(\mathcal{R}_{l-1}^{\text{feat}} \oplus \mathcal{R}_{l-1}^{\text{feat}} \times E^{\text{LiDAR}} \oplus \mathcal{R}_{l-1}^{\text{feat}} \times E^{\text{Image}}\right) \tag{9}$$

The ref-points $\mathcal{R}_L$ outputted from the MMT block are then treated as the final descriptors $\mathcal{R}$, providing multi-modal condensed environment information.

### 3.3 DPM Decoder

Following DeepPointMap [62], we use the DPM Decoder to solve the relative transformation matrix $T$ for odometry and overlap probability $p_{\text{overlap}}$ for loop detection between frames $t_1$ and $t_2$ based on their descriptors $\mathcal{R}_{t_1}$ and $\mathcal{R}_{t_2}$. In detail, the DPM Decoder contains a transformer-based block and three individual heads, as shown in Fig. 2 (b). The block exchanges information between two set of descriptors and output *correlated descriptors*, denoted as $\bar{\mathcal{R}}_{t_1}$ and $\bar{\mathcal{R}}_{t_2}$.

For odometry, the **Similarity Head** estimates the correspondence $\sigma$ between two sets of descriptors based on pairwise descriptor feature similarity. To tackle the problem led by the spatial sparsity of descriptors, **Offset Head** predict the relative offsets $\delta$ between descriptor pairs. Finally, the precise relative transformation $T$ can be estimated using weighted-SVD [2].

For loop detection, we use **Overlap Head** to predict the loop-probability $p_o$ that the distance between two frames is less than a predefined threshold $\varepsilon_{\text{loop}}$. We obtain frame-wise features for both frames by average pooling $\bar{\mathcal{R}}_{t_1}$ and $\bar{\mathcal{R}}_{t_2}$, then concatenating them and predicted the desired probability $p_o$ via an MLP.

### 3.4 Mapping

We utilize *Pose-Graph* to store our reconstructed map, following Zhang et al. [62]. For each frame, we extract its descriptors (Sec. 3.2) and retrieve its nearest keyframe from the pose-graph to estimate its pose (Sec. 3.3). A key-frame selection process then determine the frame should be assigned as a keyframe. If a frame is assigned as a

keyframe, a scan-to-map refinement will be applied to improve the pose estimation accuracy. We also conduct loop detection once a keyframe is established. A standard pose-graph optimization will be applied once the loop closure is conformed, to ensure the global consistency of the reconstructed map.

## 4 TRAINING

We jointly train the DPM Encoder and DPM Decoder end-to-end, with the following multiple losses.

**Pairing Loss.** Representing the geometry and texture features of ref-points is the key to descriptors. The ideal descriptors should share similar features if they are close in global coordinates and vice versa. Thus, we adopt InfoNCE [17] loss as pairing loss $\mathcal{L}_p$ on descriptors $\mathcal{R}$. For each descriptor $\mathbf{r}_{(i,t_1)} \in \mathcal{R}_{t_1}$ from frame $t_1$, a descriptor $\mathbf{r}_{(j,t_2)} \in \mathcal{R}_{t_2}$ from frame $t_2$ with the distance of $d_{ij} = \left\| \mathbf{r}_{(i,t_1)}^{\text{xyz}} - \mathbf{r}_{(j,t_2)}^{\text{xyz}} \right\|_2$ are assigned as (1) *positive pair* $\mathcal{R}_{t_2}^+$ iff $j = \arg\min_j d_{ij}$ and $d_{ij} \leq \varepsilon_{\text{pair}}$ where $\varepsilon_{\text{pair}}$ is a pre-defined threshold, or (2) *nature pair* $\mathcal{R}_{t_2}^{\circ}$ if $d_{ij} \leq \varepsilon_{\text{pair}}$, otherwise (3) *negative pair* $\mathcal{R}_{t_2}^-$. The pairing loss is calculated as:

$$\mathcal{L}_p = \mathbb{E}_{\mathbf{r}_{(i,t_1)}} \left[ -\log\left( \frac{\sum_{\mathbf{r} \in \mathcal{R}_{t_2}^+} \exp\left( \frac{\mathbf{r}_{(i,t_1)}^{\text{feat}} \odot \mathbf{r}^{\text{feat}}}{\tau} \right)}{\sum_{\mathbf{r} \in \mathcal{R}_{t_2}^+ \cup \mathcal{R}_{t_2}^-} \exp\left( \frac{\mathbf{r}_{(i,t_1)}^{\text{feat}} \odot \mathbf{r}^{\text{feat}}}{\tau} \right)} \right) \right] \tag{10}$$

where $\tau$ is a pre-defined constant. Note that the *nature pairs* are not contributed to this loss.

To accelerate convergence, we apply the same loss to the *correlated descriptors* $\bar{\mathcal{R}}$ as well as key-points $\mathcal{P}$, denoted as *Coarse Pairing Loss* $\mathcal{L}_c$ and *Backbone Auxiliary Loss* $\mathcal{L}_b$.

**Offset Loss.** Following the definition of three pair types above but with a different threshold $\varepsilon_{\text{offset}}$, we use both *positive* and *nurture* pairs of *correlated descriptors* $\bar{\mathcal{R}}$ to train the Offset Head to predict the offsets.

$$\mathcal{L}_o = \mathbb{E}_{\mathbf{r}_i} \left[ \frac{1}{\left| \bar{\mathcal{R}}_{t_2}^+ \cup \bar{\mathcal{R}}_{t_2}^{\circ} \right|} \sum_{\bar{\mathcal{R}}_{t_2}^+ \cup \bar{\mathcal{R}}_{t_2}^{\circ}} \left\| \delta_{i,j} - \delta_{i,j}^* \right\|_{\Sigma} \right] \tag{11}$$

where $\delta_{i,j}$ represents the predicted offset from $\bar{\mathbf{r}}_{(i,t_1)}$ to $\bar{\mathbf{r}}_{(j,t_2)}$ in $t_1$ coordinate system, and $\delta_i^*$ is its ground-truth. $\|\cdot\|_{\Sigma}$ represents the Mahalanobis distance. We utilize both *positive* and *neutral* pairs with a different distance threshold $\varepsilon_o$ to accelerate the convergence and improve the robustness.

**Overlap Loss.** We use Binary Cross Entropy (BCE) loss $\mathcal{L}_d$ to train the Overlap Head.

**Training Procedure.** We utilize two-phase training procedure discussed in Zhang et al. [62]. Phase one aims to train the registration ability of our method. We randomly sample frame pairs within 20 m from dataset, and use the loss $\mathcal{L} = \lambda_p \mathcal{L}_p + \lambda_c \mathcal{L}_c + \lambda_b \mathcal{L}_b + \lambda_o \mathcal{L}_o$ to train *DeepPointMap2*. Phase two aims to train the loop-detection ability. Thus we randomly sample frame pairs with a distance less/greater than $\varepsilon_{\text{loop}}$ with equal probability. In this phase, only the Overlap Head is trained with the loss of $\mathcal{L}_d$ whereas other modules are frozen.

**Table 1: Localization Accuracy on KITTI Odometry Benchmark (Trans↓ and Rot↓).**

| | | 06 | | 07 | | 08 | | 09 | | 10 | |
|---|---|---|---|---|---|---|---|---|---|---|---|
| **Modality** | **Method** | Trans | Rot | Trans | Rot | Trans | Rot | Trans | Rot | Trans | Rot |
| **LiDAR** | LOAM [60] | 0.65 | - | 0.63 | - | 1.12 | - | 0.77 | - | 0.79 | - |
| | LO-Net [22] | - | - | 0.56 | 0.45 | 1.08 | 0.43 | 0.77 | 0.38 | 0.92 | 0.41 |
| | ISC-LOAM [53] | 0.76 | 0.41 | 0.56 | 0.43 | 1.20 | 0.50 | 1.40 | 0.59 | 1.87 | 0.62 |
| | SC-LeGO-LOAM [19] | 2.54 | 1.15 | 2.48 | 1.78 | 2.30 | 1.24 | 5.37 | 2.78 | 10.50 | 3.79 |
| | F-LOAM [52] | 0.84 | 0.33 | 0.88 | 0.62 | 0.87 | 0.33 | 1.03 | 0.32 | 1.20 | 0.29 |
| | LiODOM [14] | 0.83 | 0.29 | 0.88 | 0.61 | 0.86 | 0.33 | 1.03 | 0.32 | 1.20 | 0.29 |
| | LiLO [48] | 0.54 | 0.32 | 0.60 | 0.61 | 1.07 | 0.41 | 0.63 | 0.32 | 0.99 | 0.33 |
| **Camera** | VISO2 [16] | 0.79 | 0.51 | 1.46 | 1.13 | 1.62 | 0.66 | 0.84 | 0.64 | 1.29 | 0.64 |
| | ORB-SLAM2 [32] | 0.89 | 0.27 | 0.89 | 0.50 | 1.03 | 0.31 | 0.86 | 0.25 | 0.62 | 0.29 |
| | VINS-Fusion [37, 38] | 1.35 | 0.71 | 1.21 | 0.90 | 1.83 | 0.72 | 1.82 | 0.53 | 2.64 | 1.01 |
| | OV2-SLAM [12] | 1.13 | 0.28 | 1.03 | 0.57 | 1.11 | 0.31 | 0.96 | 0.20 | 0.52 | 0.18 |
| | SOFT2 [7] | 0.60 | 0.23 | 0.45 | 0.29 | 0.91 | 0.26 | 0.75 | **0.22** | 0.74 | **0.24** |
| **LiDAR+Camera** | DEMO [57] | 0.96 | - | 1.16 | - | 1.24 | - | 1.17 | - | 1.14 | - |
| | DVL-SLAM [43] | 0.92 | - | 1.26 | - | 1.32 | - | 0.66 | - | 0.70 | - |
| | Huang et al. [18] | 0.61 | - | 0.56 | - | 1.27 | - | 1.06 | - | 0.83 | - |
| | LAMV-SLAM [55] | 0.49 | - | 0.84 | - | 1.19 | - | 0.80 | - | **0.55** | - |
| | *DeepPointMap2* | **0.47** | **0.20** | **0.39** | **0.25** | **0.77** | **0.22** | **0.62** | 0.23 | 0.75 | 0.40 |

## 5 EXPERIMENTAL ANALYSIS

### 5.1 Settings

**Datasets.** Our experiments utilize three multi-modal autonomous driving-oriented datasets: (1) The KITTI Odometry Dataset [15], a widely used benchmark containing 11 LiDAR-Camera sequences (00-10), encompassing diverse scenarios from urban to highway environments. (2) KITTI-360 [24], a large dataset with 9 LiDAR-Camera sequences that introduce challenges with longer distances and more complex environment. (3) KITTI-Carla [8], a simulated dataset with 6 noise-free LiDAR-Camera sequences generated by Carla [9] simulator, which are used to assist training.

**Settings.** The model is trained on 6× RTX 3090 GPUs, with AdamW optimizer [40], initial $lr = 1 \times 10^{-3}$, $wd = 1 \times 10^{-4}$, and cosine scheduler. The training set contains the first 6 sequences of KITTI Odometry dataset (00-05), the first 6 sequences of KITTI-360 dataset (00, 02-06), and the entire KITTI-Carla dataset (Town01-06). Since the ground-truth label in the original KITTI Odometry is not ideal, we utilize a more precise label provided in SemanticKITTI [3] to train our model. However, we still use the original KITTI ground-truth for evaluation to make a fair comparison. The evaluation model is trained for 21 epochs for Phase One and another 10 for Phase Two. During Phase Two the $lr$ and $wd$ are decayed with a rate of 0.1. We set the loss weight $\lambda_p, \lambda_c, \lambda_b, \lambda_o = 1, 0.1, 0.1, 1$, the threshold $\varepsilon_{pair} = 1$m, $\varepsilon_{offset} = 2$m and $\varepsilon_{loop} = 20$m.

**Metrics.** We adopt the official metrics of each benchmark for quantitative evaluation: We use *Relative Translation Error* (Trans↓) (%) and *Average Rotation Error* (Rot↓) (°/100m) to measure relative localization accuracy in KITTI Odometry benchmark, and use the *Mean Absolute Pose Error* (APE↓) (m) to evaluate the global trajectory accuracy for KITTI-360.

### 5.2 Localization Accuracy

This experiment aims to demostrate the localization accuracy of our proposed method, *DeepPointMap2*, in various road scenarios. The experiment is conducted on five KITTI Odometry sequences (06-10) with comparison methods divided into LiDAR-based, Visual-Based and LiDAR-Visual-based groups, where each group contains multiple widely-used and advanced SLAM approaches.

As presented in Tab. 1, *DeepPointMap2* achieves the lowest translation error in four sequences and the lowest rotation error in three sequences. Some methods do not report rotation error metrics, which are marked with "-". Sequence 08 is the longest sequence among them, covering a larger urban area and containing multiple loops. In such complex scenario, *DeepPointMap2* significantly outperforms all LiDAR-Visual-based methods, reducing translation error by 35% compared to the existing SOTA method LAMV-SLAM [55].

To illustrate the superiority of our proposed method in large-scale scenes, we select two representative sequences (07 and 09) from the KITTI-360 benchmark, and compare *DeepPointMap2* with the recent SOTA LiDAR-based method DeepPointMap. Sequence 07 was collected in a highway/urban roadway with a length of 4.9 km. The absence of loop closures in this sequence presents a challenge for odometry. Sequence 09, on the other hand, was collected in a complex, large-scale urban environment, with a trajectory length of over 10.5 km. The environmental complexity and numerous loops challenge the model's loop closure capability.

As shown in Fig. 5, our *DeepPointMap2* demonstrates an advantage in global trajectory estimation error by achieving the ATE of 26.00 in sequence 07, which is better than the APE of 93.77 achieved by DeepPointMap. The challenge in the highway scenario lies in the monotonous geometric patterns, which make accurate odometry difficult when relying solely on LiDAR point clouds. However, the inclusion of the visual modality, with its rich textural information,

**Table 2: Robustness of *DeepPointMap2* when Camera Unavailable and LiDAR Obstruction (Trans↓) and Rot↓**

| Scenario | Frame% | 06 Trans | 06 Rot | 07 Trans | 07 Rot | 08 Trans | 08 Rot | 09 Trans | 09 Rot | 10 Trans | 10 Rot | Mean Trans | Mean Rot |
|---|---|---|---|---|---|---|---|---|---|---|---|---|---|
| **Camera** | 5% | 0.50 | 0.24 | 0.38 | 0.22 | 0.76 | 0.21 | 0.70 | 0.27 | 0.76 | 0.42 | 0.62 +3% | 0.27 +5% |
| **Failure** | 30% | 0.46 | 0.20 | 0.36 | 0.21 | 0.86 | 0.26 | 0.66 | 0.27 | 0.71 | 0.28 | 0.61 +2% | 0.24 -6% |
|  | 50% | 0.53 | 0.25 | 0.47 | 0.30 | 0.88 | 0.28 | 0.85 | 0.35 | 0.72 | 0.38 | 0.69 +15% | 0.31 +20% |
|  | 100% | 0.64 | 0.30 | 0.50 | 0.33 | 1.46 | 0.44 | 1.76 | 0.51 | 0.83 | 0.49 | 1.04 +73% | 0.41 +59% |
| **LiDAR** | 5% | 0.44 | 0.19 | 0.37 | 0.22 | 0.81 | 0.23 | 0.70 | 0.26 | 0.80 | 0.35 | 0.62 +4% | 0.25 -4% |
| **Obstruction** | 30% | 0.43 | 0.17 | 0.47 | 0.30 | 0.96 | 0.24 | 0.81 | 0.27 | 0.64 | 0.32 | 0.66 +10% | 0.26 +0% |
|  | 50% | 0.40 | 0.17 | 0.55 | 0.35 | 0.92 | 0.26 | 0.85 | 0.29 | 0.85 | 0.39 | 0.71 +19% | 0.29 +12% |
|  | 100% | 0.37 | 0.16 | 0.53 | 0.40 | 1.03 | 0.34 | 0.86 | 0.29 | 0.89 | 0.37 | 0.74 +23% | 0.31 +20% |
| **Normal Input** |  | 0.47 | 0.20 | 0.39 | 0.25 | 0.77 | 0.22 | 0.62 | 0.23 | 0.75 | 0.40 | 0.60 | 0.26 |

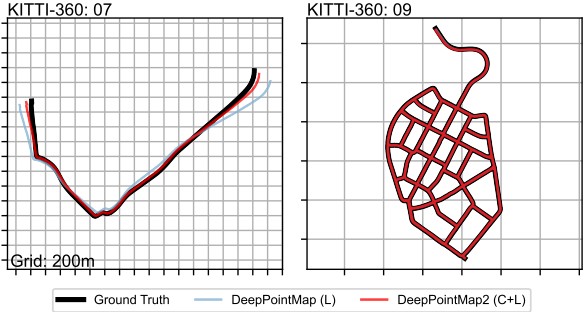

**Figure 5: Estimated Trajectories on KITTI-360 Benchmark.**

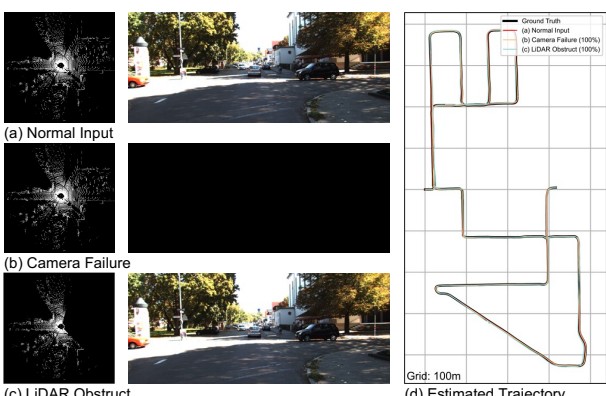

**Figure 6: Robustness of *DeepPointMap2*.**

enhances the representation ability and, consequently, improves odometry accuracy in such scenarios. As for sequence 09, our method successfully achieves accurate localization and constructs a precise map in large-scale complex urban environments, showcasing its robustness and adaptability.

### 5.3 Strong Robustness

In certain real-world scenarios, one of the sensors (*i.e.*, LiDAR or camera) may be temporarily unavailable or experience a degradation in performance, posing significant challenges for multi-modal SLAM models. To investigate the robustness of our approach, we design two additional experiments to evaluate our multi-modal *DeepPointMap2* under these conditions, without any finetuning.

In the first experiment, we simulate camera failure scenarios by randomly selecting x% (x=5,30,50,100) of frames and blacking out their image input, as shown in Fig. 6 (b). Quantitative results indicate that *DeepPointMap2* consistently produces remarkable results as image modalities are missing, as shown in Tab. 2. Even when 50% of the images are absent, our approach incurs only an average performance decrement of approximately 15%. Furthermore, when all image modalities are unavailable, the model still functions normally in most scenes (*e.g.*, sequences 06, 07 and 10) and demonstrates competitive performance with other methods on sequence 08 and 09. Meanwhile, we also evaluate the robustness of our method in scenarios where the LiDAR point cloud is obstructed by large vehicles (*e.g.*,

trucks or buses), as demonstrated in Fig. 6 (c). To simulate partial obstruction, we randomly select x% of frames and perform the following steps: (1) generate a random-sized and -oriented 3D box at a random location, and (2) remove all LiDAR points that pass through this box. Despite this challenge, our *DeepPointMap2* successfully maintains robust and consistent localization performance. This is attributed to the method's ability to extract discernible features and the adaptability of the cross-modal fusion process, which is further enhanced by the implementation of the *RandomOcclusion* data augmentation technique Even in situations where all frames are affected by occlusion, our method's performance exhibits only a modest average decline of approximately 23%, and it remains competitive when compared to other SOTA Visual-LiDAR-based methods, as evidenced by the quantitative analysis presented in Tab. 2. The visualization of the localization and mapping result on KITTI sequence 08 with the LiDAR obstruction rate and camera failure rate of 100% are shown in Fig. 6 (right). It can be observed that under such harsh scenarios, *DeepPointMap2* still successfully reconstructed the challenging scenes.

### 5.4 Ablation Study

**Reference-Point Generation.** As mentioned in Sec. 3.2.1, we use *farthest-point-sample* strategy to sample $n = 256$ reference-points $\mathbf{r}^{xyz}$ from the input point cloud. In addition, we adapt *uniform-sampling* and *normal-sampling* methods to generate ref-points with

**Table 3: Reference Point Generate Strategies (Trans↓).**

| Dist. | Num. | 06 | 07 | 08 | 09 | 10 |
|---|---|---|---|---|---|---|
| FPS | 128 | 0.49 | **0.38** | 0.94 | 0.85 | 0.78 |
| | 256 | **0.47** | 0.39 | **0.77** | **0.62** | **0.75** |
| | 512 | 0.57 | 0.41 | 0.88 | 0.74 | 0.85 |
| Uniform | 256 | 25.97 | 31.91 | 13.48 | 13.50 | 13.98 |
| | 4096 | 6.92 | 6.25 | 10.79 | 9.86 | 16.58 |
| Normal | 256 | 25.19 | 5.21 | 11.06 | 26.71 | 20.85 |
| | 4096 | 5.83 | 39.70 | 18.04 | 14.88 | 71.25 |

different numbers. The *uniform-sampling* variant of the RPG module produces $n$ points drawn from the uniform distribution $\mathbf{r}^{xyz} \sim \mathcal{U}_{[0,1]}$ and scales them linearly to fit the 3D world space. The *normal-sampling* variant follows a similar procedure but generates points from a normal distribution $\mathbf{r}^{xyz} \sim \mathcal{N}(0, 0.25)$.

As indicated in Tab. 3, the localization performance peaks with $n = 256$ ref-points when employing the FPS strategy. Reducing the ref-points number can result in a greater distance between matched descriptors, which may reduce the model's ability to accurately predict the offset values $\delta$. Conversely, increasing the number of ref-points to 512 does not yield a significant enhancement in performance, as the overly close ref-points fail to capture distinct features from the key-points, given that the LiDAR backbone only extracts 256 key-points.

For alternative sampling strategies such as random distribution, their main shortcoming stems from the extensive nature of the point cloud. Many ref-points are generated far from any points in the point cloud (and key points), which poses a challenge for the network to effectively aggregate features for these remote points. Furthermore, the vast scale of the scene leads to a sparse arrangement of ref-points, potentially surpassing the maximum range (*i.e.*, $\varepsilon_{offset}$) within which Overlap Head can operate optimally. This scenario compromises the effectiveness of the offset compensation mechanism, thus diminishing overall performance. Due to the aforementioned factors, neither uniform nor normal distributed sampling can construct a reasonable map at $n = 256$. It is only when $n$ is increased to 4096 that the model can achieve minimal localization accuracy and build recognizable maps in benchmark sequences.

**Fusion Module Design.** In this additional experiment, we explore the importance and advantages of our proposed Biased Transformer. Some existing methods such as PointPainting [51] focus on pixel-level early-fusion, where each LiDAR point is associated with a pixel and the RGB values are attached to the point before processing by the LiDAR backbone. In contrast, other multi-modal models opt for feature-level fusion, integrating features from both image and Li-DAR modalities. As the most straightforward approach, RoI-Pooling aggregates the feature within a window and attaches these pooled features to the corresponding points. However, the fixed window size results in a lack of scale invariance, where the pooling region *should but not* appears smaller at a distance and larger when close to the point.

Some attention-based methods can also be used for cross-modal fusion. DeepFusion [23], as one of the SOTA 3D detectors, employs a learnable scale-dot cross-attention module to fuse LiDAR and camera features. Our Biased Transformer introduces two branches that

**Table 4: Attention Module Design (Trans↓).**

| Fusion Strategy | 06 | 07 | 08 | 09 | 10 |
|---|---|---|---|---|---|
| Point Painting | 0.56 | 0.56 | 1.15 | 0.94 | 1.55 |
| RoI Pooling (3×3) | 2.34 | 1.56 | 4.12 | 4.24 | 17.83 |
| Scale-Dot Attention | 2.73 | 1.00 | 9.63 | 6.48 | 6.48 |
| *w/o Learnable-Attn.* | 0.72 | 0.46 | 18.00 | 1.16 | 1.21 |
| *w/o Spatial-Attn.* | 41.54 | 21.54 | 29.91 | 26.28 | 20.87 |
| Biased Attention (ours) | **0.47** | **0.39** | **0.77** | **0.62** | **0.75** |

leverage both feature similarity and spatial distance. In this experiment, we disable each branch to investigate its importance. To save the computation, we follow Yin et al. [56] and fine-tune all the ablation models (based on the modal evaluated in Sec. 5.2) on the KITTI dataset for 10 epochs.

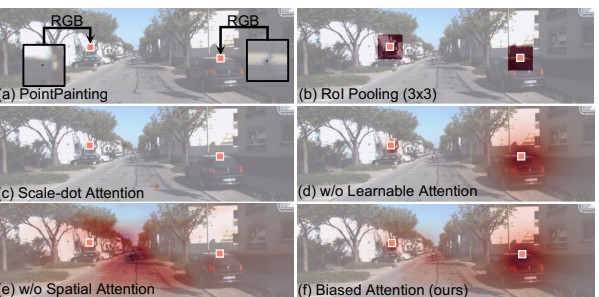

**Figure 7: Attention Map Visualization.**

As observed in Tab. 4 and Fig. 7, the performance decreased when replacing the fusion strategy. Although attention-free approaches (*e.g.*, Painting and RoI) have advantages in inference speed, both approaches exhibit weaknesses in localization performance. By removing the *learnable-attention map*, the model relies on the prior Camera-LiDAR calibration and still maintains a relatively high localization accuracy, except for one loop detection error in seq08. Removing the *Spatial-Attention Map* brings difficulty of learning the correspondence between images and point clouds, resulting in failure to localize and map in most of the sequences. Finally, the classic scale-dot attention module successfully reconstructs most of the sequences. However, since the spatial information only exists in the positional embeddings, the model is required to learn an appropriate spatial attention map by itself, resulting in slower convergence and reduced performance, compared to Biased Attention.

## 6 CONCLUSIONS

We present *DeepPointMap2*, a novel learning-based LiDAR-Visual SLAM architecture that leverages a flexible *Visual-Point Fusion Module*. This module adeptly aggregate multi-modal tokens, ensuring precise and resilient performance even in adverse conditions such as LiDAR obstructions and camera failures.

**Limitation.** The RPG module utilizes a parameter-free strategy, FPS, to obtain initial ref-point. We believe that a learn-based strategy can be used to actively sample these ref-points and avoid sampling points from dynamic objects (*e.g.*, cars).

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
