# OpenReview forum: "DeepPointMap2: Accurate and Robust LiDAR-Visual SLAM with Neural Descriptors"
_acmmm.org/ACMMM/2024/Conference — MM2024 Poster_

### Official Review · Reviewer_p1Rc · 2024-05-19

**Rating:** 4
**Confidence:** 3

**Summary:**

DeepPointMap2 is a LiDAR-Visual SLAM system that uses neural descriptors for robust and accurate localization in autonomous driving and robotics. It overcomes the limitations of traditional methods by employing neural networks to extract multi-modal features and fuse them into 3D descriptors, achieving top performance in benchmarks and maintaining robustness in challenging scenarios like camera failure or LiDAR obstruction.

**Strengths:**

1. DeepPointMap2 introduces an innovative, learning-based architecture within the domain of multi-modal SLAM, showcasing significant innovation. It leverages the novel Visual-Point Fusion Module to seamlessly amalgamate feature tokens derived from diverse data modalities, culminating in the generation of sophisticated sparse neural 3D descriptors. This approach guarantees accurate localization and delivers a high level of robust performance.
2. The manuscript exhibits a well-organized structure and a coherent logical flow. The inclusion of diagrams and visual representations significantly enhances the comprehension of the methodologies presented. The experimental setup is meticulously designed, and the outcomes robustly substantiate the efficacy of the proposed techniques.

**Limitations:**

1. DeepPointMap2 emerges as an innovative framework that builds upon the foundation laid by DeepPointMap. Nonetheless, during a comparative analysis of the methodologies, Section 3 detailing the Architecture reveals that DeepPointMap2 primarily incorporates additional Visual image inputs and debuts the Visual-Point Fusion Module designed to fuse the LiDAR and Visual features. It is observed that the remainder of the system closely mirrors DeepPointMap, thus warranting a more explicit delineation of the distinctions between the two iterations.
2. In the exposition of Section 4 on Training, the approach appears to align entirely with that of DeepPointMap. It would be beneficial to clarify if there are any divergent aspects.
3. Furthermore, it is noted with curiosity the absence of a comparative analysis with the DeepPointMap methodology within the experimental section, specifically Table 1.
4. There are issues with the presentation of some formulas in Section 3. For instance, in Eq. 5, T represents the transformation matrix from the camera to LiDAR, which is typically a 4x4 matrix, and the intrinsic matrix of the camera, K, is a 3x3 matrix. It is unclear how these two matrices are multiplied. In Eq. 9, the symbols “⊕” and “×” are used, but their meanings are not explained. It is recommended that the author reviews such errors to enhance the readability of the paper.

**Suitability:**

2

---

### Official Review · Reviewer_gALx · 2024-05-20

**Rating:** 4
**Confidence:** 3

**Summary:**

The paper introduces DeepPointMap2, an advancement over the previous work [1], focusing on fusing image and point cloud data for Simultaneous Localization and Mapping (SLAM). DeepPointMap2 employs neural descriptors for multi-modal (image and point cloud) feature extraction and fusion, enhancing the robustness and accuracy of SLAM systems. This approach leverages neural networks to adaptively combine visual and LiDAR data, generating robust 3D descriptors for precise localization. Experimental results demonstrate that DeepPointMap2 achieves state-of-the-art performance across various benchmarks, maintaining high accuracy even in challenging scenarios such as sensor failure or obstruction.

[1] X. Zhang, Z. Ding, Q. Jing, Y. Zhang, W. Ding, and R. Feng, ‘DeepPointMap: Advancing LiDAR SLAM with Unified Neural Descriptors’, in Proceedings of the AAAI Conference on Artificial Intelligence, 2024, vol. 38, pp. 10413–10421.

**Strengths:**

The paper is well-written and easy to read. The strength of this work is highlighted below:

-**Innovative Fusion Strategy**: DeepPointMap2 introduces an adaptive fusion strategy that combines visual and LiDAR data using neural descriptors, enhancing feature representation and robustness.

-**State-of-the-Art Performance**: The approach achieves superior localization accuracy on widely used benchmarks, demonstrating its effectiveness in various environments.

-**Robustness**: The system shows strong resilience in challenging scenarios, such as LiDAR obstruction and camera failure, maintaining high localization accuracy.

**Limitations:**

My primary concerns about this work revolve around the robustness evaluation.

1. Randomly blacking out images does not accurately simulate real-world sensor failures, as it is uncommon for cameras to fail intermittently. In reality, a camera failure would result in consecutive frames being blacked out for a short or long period, depending on whether the camera can recover.

2. The results presented in Table 2 are insufficient to demonstrate the impact of the proposed method. Including a comparison with existing work would better highlight the robustness of the proposed method, as it is difficult to convincingly argue that a 15% drop in performance is insignificant without a proper benchmark comparison.

Other weaknesses:

-Increased Complexity: The integration of multiple neural network modules and fusion strategies adds to the system's complexity, potentially affecting real-time performance.

**Suitability:**

3

---

### Official Review · Reviewer_UCRm · 2024-05-22

**Rating:** 3
**Confidence:** 2

**Summary:**

This paper introduces a learning-based LiDAR-Visual SLAM approach, which integrates features from both images and point clouds to enhance location accuracy and robustness.

**Strengths:**

1. This paper introduces multi-modal neural 3D descriptors to represent the environment, which have the potential to be utilized in any single-modal backbone.
2. This paper proposes an end-to-end learnable framework to adaptively learn how to fuse multi-modal features, providing better feature representation and robustness. It is stimulating to the MM community.
3. This paper is well-organized, including the tables, figures, and equations.

**Limitations:**

1. The main contribution of this paper is the LiDAR-Visual fusion methodology. However, the comparison with existing fusion methods in Table 1 seems to be out-of-date. For example, the latest fusion method compared in this paper is LAMV-SLAM (2022).
2. The comparison to DeepPointMap is insufficient. Only lines 689-695 demonstrate the comparison in the scene "sequence 07." Could you provide more comparisons with the previous work DeepPointMap to better distinguish the contributions from DeepPointMap? This would be similar to how you compare with other LiDAR methods in Table 1.
3. In Table 2, this paper presents experiments to show the robustness of the method. However, the robustness seems to indicate that the model still works when either the camera or the LiDAR is missing. I wonder, how will it perform when 5% of the image frames and 5% of the LiDAR frames are missing in the same sequence?

**Suitability:**

3

---

### Official Review · Reviewer_6Bko · 2024-05-22

**Rating:** 3
**Confidence:** 3

**Summary:**

This paper focuses on multi-modal SLAM systems and targets the insufficient flexibility and robustness of hand-craft features. The paper presents a learning-based LiDAR-Visual SLAM and leverages neural networks to enhance the feature extraction and localization processes in SLAM tasks. In particular, separate backbones are used to extract feature tokens from both image and lidar point clouds, which are then aggregated into 3D neural descriptors. Then the descriptors are used for pose estimation and loop closure detection. The proposed SLAM system is evaluated on KITTI, KITTI-360, KITTI-Carla datasets, and achieves better performance compared to other visual, lidar, and lidar-visual SLAM systems.

**Strengths:**

- the paper presents a visual point fusion module to fuse features extracted from images and lidar point clouds. The procedure is clearly explained and technically sound.

- the proposed learning-based SLAM demonstrated superior performance on KITTI datasets, and extensive ablation studies were conducted to demonstrate the effectiveness of the proposed design.

- the proposed learning-based SLAM exhibits strong robustness when the input of the image modality or the Lidar modality is corrupted.

**Limitations:**

- in line 389, the authors stated the biased transformer module is illustrated in Fig. 3 (c), while figure 3 in the paper only has 2 sub figures, (a) and (b).

- eq 9, what is the meaning of ⊕? Looks like concatenation as indicated in the figure, but why different symbols are used?

- in section 3.4, More details are recommended, if a new keyframe is assigned, scan-to-map refinement is applied. But the paper didn’t mention what form the reconstructed map is stored. I assume it is in the point cloud?

- in section 4, infonce loss is also applied to the correlated descriptor and the key points.
  1) how exactly are the correlated descriptors generated? Section 3.3 stated that the correlated descriptors are outputs of the DPM decoder, but the only outputs shown in Fig. 2 are pose and overlap probability.
  2) the author stated the reason to apply infonce loss on correlated descriptors and keypoints is to accelerate convergence, any ablation studies to verify this? I wonder whether it introduces performance improvements.

- in section 5.1, more details and explanations are recommended.
  1) the choice of using ground truth label from SemanticKITTI for training and use the label from KITTI for evaluation is questionable. Why not directly evaluate on SemanticKITTI and what is the performance difference using KITTI label for training?
  2) the model is trained for 21 epochs in phase one, which is not a usual choice. More explanations on this choice would be preferred.

- Also, the proposed learning-based SLAM is trained and evaluated on KITTI/KITTI-360 datasets. i.e. the training and testing scenes are very similar, which raises the concerns of overfitting.

- all the quantitative results are from KITTI dataset. Since KITTI-360 is also evaluated, it would be interesting to see some more quantitative comparisons on KITTI-360. Also, in lines 690 and 691, ATE reported for DeepPointMap2 and APE reported for DeepPointMap, typo?

- section 5.3, it would be interesting to see some discussion on why the performance even improves in the scenario of lidar obstruction (eg. seq 06).

- section 5.4, it would be better to move the choice of reference point number to the setting section.

minor: typo in line 556. nurture -> nature, line 566 neutral -> nature?

**Suitability:**

3

---

### Meta-Review · Area_Chair_vare · 2024-07-03

**Recommendation:** Accept (Poster)
**Confidence:** 4

**Metareview:**

The paper proposes a learning-based LiDAR-Visual SLAM architecture that leverages neural descriptors to tackle multiple SLAM subtasks in a unified manner. The method achieves good localization performance among various Visual-based, LiDAR-based, and Visual-LiDAR-based methods in widely used benchmarks.

All reviewers agree on the suitability of the work for ACMMM, the method's novelty and robustness.